# Three-gene prognostic biomarkers for seminoma identified by weighted gene co-expression network analysis

**Hualin Chen, Gang Chen**[ORCID]**\*, Yang Pan, Xiaoxiang Jin**

Department of Urology, The First Affiliated Hospital of Chongqing Medical University, Chongqing, China

\* chengang2308@163.com

## Abstract

Testicular germ cell tumors (TGCTs) are common in young males, and seminoma accounts for a large proportion of TGCTs. However, there are limited records on the exploration of novel biomarkers for seminoma. Hence, we aimed to identify new biomarkers associated with overall survival in seminoma. mRNA-seq and clinical traits of TGCTs were downloaded from UCSC XENA and analyzed by weighted gene co-expression network analysis. After intersection with differentially expressed genes in GSE8607, common genes were subjected to protein-protein interaction (PPI) network construction and enrichment analyses. Then, the top 10 common genes were investigated by Kaplan–Meier (KM) survival analyses and univariate Cox regression analyses. Ultimately, *TYROBP*, *CD68*, and *ITGAM* were considered three prognostic biomarkers in seminoma. Based on correlation analysis between these genes and immune infiltrates, we suggest that the three biomarkers influence the survival of seminoma patients, possibly through regulating the infiltration of immune cells. In conclusion, our study demonstrated that *TYROBP*, *CD68*, and *ITGAM* could be regarded as prognostic biomarkers and therapeutic targets for seminoma patients.

**Data Availability Statement:** The analyzed GEO dataset with 40 samples of seminoma and 3 samples of controls, can be found with accession number GSE8607 (https://www.ncbi.nlm.nih.gov/geo/query/acc.cgi?acc=GSE8607).

## Introduction

Testicular germ cell tumors (TGCTs) are the most common malignancy in males between the ages of 15 and 35 years [1]. According to the GLOBOCAN database, there are approximately 71,000 new cases and 9500 deaths from TGCTs per year, worldwide [2]. Pathological studies reveal that seminoma accounts for over 60% of TGCT cases and this proportion is increasing [3]. Additionally, approximately 80% of seminomas are classified as stage I according to the clinical staging system [4]. Although the primary treatment for seminoma can result in a 5-year survival rate of over 90%, some patients fail cisplatin-based first-line chemotherapy and about 3%–5% of them will eventually die of the disease [5]. Moreover, for patients with solitary testicle or bilateral testicular tumors, novel treatment methods are needed to increase survival rate. Recently, from in-depth studies surrounding tumor immunity, immunotherapy has become a potential therapeutic method for patients with seminoma [5]. Hence, it is essential to identify novel biomarkers and understand the molecular mechanism of tumorigenesis with

**Funding:** Corresponding author GC received the award Grant number: cstc2015shmszx120067 Full name of the funder: Chongqing Science and Technology Commission URL of the funder website: http://kjj.cq.gov.cn/ The funders had no role in study design, data collection and analysis, decision to publish, or preparation of the manuscript.

**Competing interests:** The authors have declared that no competing interests exist.

an attempt to obtain early diagnosis, better clinical application of novel treatment strategies, and prognostic prediction.

In recent years, the rapid development of microarray technologies and high-throughput sequencing technologies has provided promising approaches for screening and identifying novel therapeutic targets and prognostic biomarkers for seminoma. Weighted gene co-expression network analysis (WGCNA), which was primarily developed by Peter Langfelder and Steve Horvath, is an advanced method for exploring the correlations between genes and clinical traits. In WGCNA, the concept of soft threshold has been raised, instead of the hard threshold used in traditional bioinformatics analysis. Therefore, potential key genes with small fold changes, which may be strongly correlated with clinical traits and play important roles in tumorigenesis, may be identified through the network [6, 7].

In the present study, we used WGCNA to identify seminoma-correlated modules and core genes in an attempt to provide novel therapeutic targets and obtain a better understanding of the molecular mechanisms driving seminoma.

## Materials and methods

### Data acquisition and pre-processing

The workflow of this study is presented in Fig 1. The transcriptome data and clinical information of TGCT were downloaded from the TCGA Hub of the UCSC XENA database (https://tcga.xenahubs.net). Pure samples of seminoma and non-seminoma (embryonal carcinoma, choriocarcinoma, yolk sac tumor and teratoma) were screened for further analysis, while samples without clinical traits and mixed samples were removed. Subsequently, 121 samples including 66 samples of seminoma and 55 samples of non-seminoma were identified. Then, genes were ranked by median absolute deviation (MAD) from high to low, and the top 5000 MAD genes were identified for co-expression network analysis.

### Construction of a weighted correlation network and identification of modules associated with seminoma

Primarily, a correlation matrix was constructed using Pearson's correlation coefficient matrices which were calculated by average linkage method for all pairwise genes. Then, the correlation matrix was transformed into a weighted adjacency matrix using the soft-thresholding function. By utilizing the soft-connectivity algorithm, a co-expression network with a balance between scale-independence and mean connectivity was obtained. Scale-independence >0.85 and average connectivity <100 were used as the criteria for a suitable soft threshold. Subsequently, the adjacency matrix was transformed into a topological overlap matrix (TOM). Dissimilarity (1-TOM) was calculated and considered as the distance measurement to cluster genes with similar expression profiles into gene modules with a minimum size cutoff of 30. A merge height of 0.25 was used as a criterion to cluster similar modules. P-values and correlation coefficients were calculated to identify the association between a co-expression module and the clinical phenotype.

After that, the blue and green modules were considered as the two hub seminoma-correlated modules. Then, preliminary Kyoto Encyclopedia of Genes and Genomes (KEGG) pathway enrichment analyses were performed for genes in these two modules to determine the more significant module. Ultimately, the blue module was identified as the candidate seminoma-correlated module since the preliminary KEGG analyses revealed no enriched pathway for genes in the green module and some interesting pathways potentially related to tumor biology for genes in the blue module. Hence, these genes were selected for further analyses. The

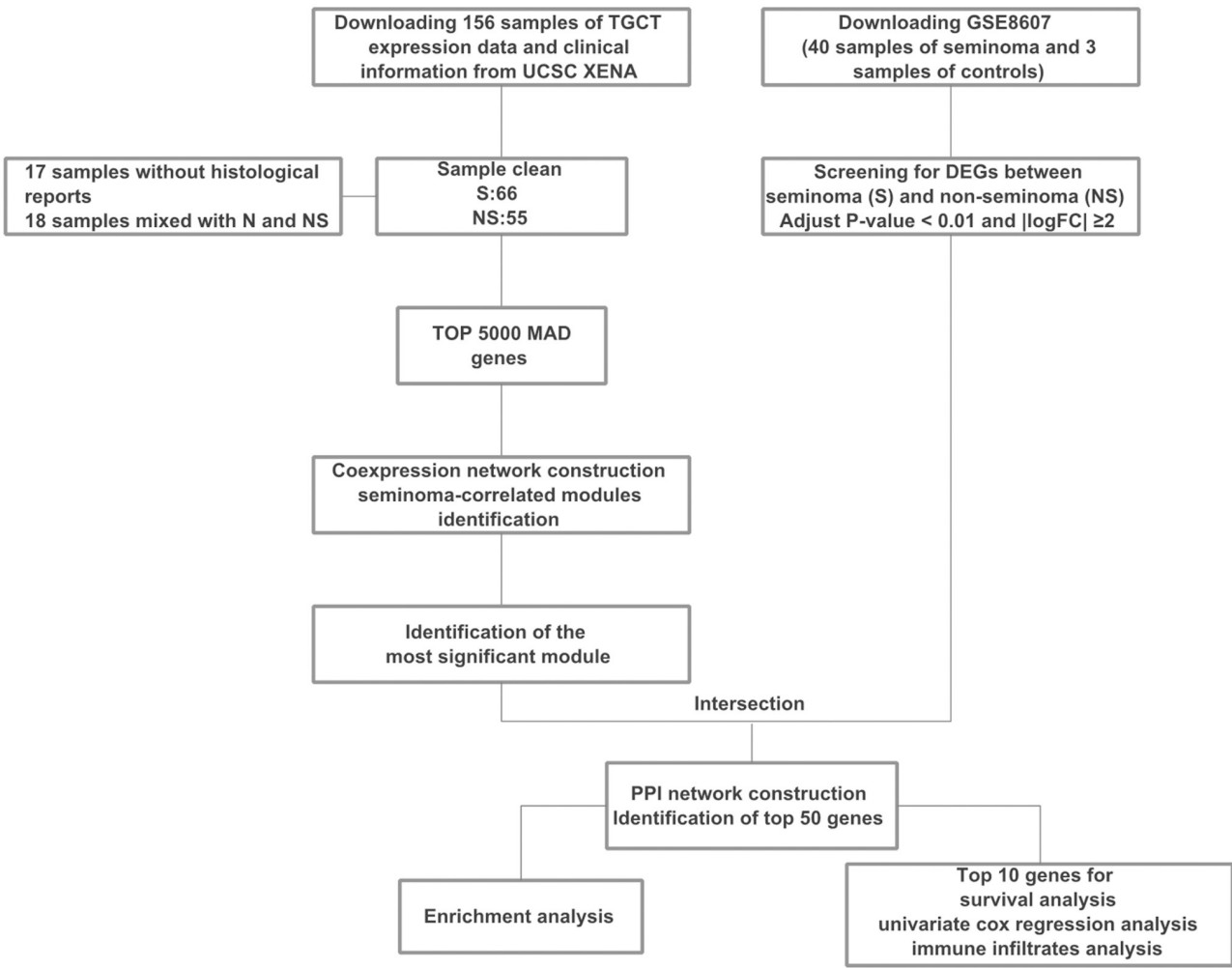

**Fig 1. Flow chart indicating the workflow used for prognostic biomarkers selection in the analysis.** TGCT, testicular germ cell tumors. MAD, median absolute deviation. PPI, protein-protein interaction.

clusterProfiler package in R was used for KEGG analyses [8]. The workflow of WGCNA is presented in S1 Fig.

## Identification of DEGs in GSE8607

We screened the DEGs between seminoma and controls in GSE8607 using the 'limma' R package [9]. Adjust P-value < 0.01 and |logFC| ≥2 were set as the cutoff criterion for improved accuracy and significance, as described previously [10]. Heatmap and volcano maps were drawn to present the DEGs.

## Identification of common genes, PPI network construction and functional annotation

Common genes in both the blue module and DEGs obtained in GSE8607 were identified using the VennDiagram package in R [11].

The common genes were submitted to The Search Tool for the Retrieval of Interacting Genes/Proteins (STRING) database for protein-protein interaction (PPI) network

construction. Then, the PPI network was downloaded and visualized with Cytoscape software [12]. Hub genes were identified using the plug-in CytoHubba [13]. With the application of betweenness centrality (BC) algorithm, the top 50 genes were identified for further functional annotation [14].

Gene Ontology (GO) and pathway functional enrichment analysis were performed using the clusterProfiler package in R and REACTOME Pathway databases (https://reactome.org), respectively. Next, the top 10 genes by highest BC were further identified for identification of hub genes.

### Identification of prognostic biomarkers

To investigate the clinical significance of the top 10 gene signatures, Kaplan–Meier survival analyses and univariate Cox regression analyses were performed based on survival data and normalized expression profiles of TGCT obtained from the UCSC XENA platform. The Human Protein Atlas was used for validating the immunohistochemistry (IHC) of latent hub genes [15, 16]. The Survival and Survminer packages in R were used to analyze and visualize the survival data and P<0.05 was considered as indicating a statistically significant difference.

After identification of significant genes, their expression values in multiple tumors were identified by consulting the GEPIA, a newly developed interactive web server for analyzing the RNAseq data of tumors and normal samples from TCGA and GTEx projects [17].

### Immune infiltrate analysis in seminoma

The Tumor Immune Estimation Resource (TIMER) database, which is a comprehensive resource for the systematic analysis of immune infiltrates across multiple tumors, was employed to analyze the correlation between prognostic biomarkers and the abundance of immune cell infiltrates, including B cells, CD4+ T cells, CD8+ T cells, Neutrophils, Macrophages, and Dendritic cells [18]. Tumor purity was calculated using R package and CHAT, as described in a previous study [19]. Subsequently, correlations between prognostic biomarker expression and gene markers of tumor-infiltrating immune cells were further explored using Spearman's correlation. Gene markers have been reported in previous studies [20].

## Results

### Construction of the co-expression network and identification of seminoma-correlated modules

As presented in Fig 2A, a power of 9 was selected as the soft threshold to construct the weighted adjacency matrix. Based on the dissimilarity of the topological overlap matrix, a cluster dendrogram was generated (Fig 2B).

As presented in Fig 2C, seminoma was significantly correlated with the blue module ($r^2 = 0.54$, $P = 10^{-10}$) and green module ($r^2 = 0.78$, $P = 5x10^{-26}$). Scatterplots of Gene Significance vs. Module Membership in the two modules were plotted (Fig 2D and 2E).

Then, preliminary KEGG enrichment analyses for genes in the two modules were performed. No term was enriched in the 407 genes in the green module, while 738 genes in the blue module were mainly enriched in pathways related to tumor growth, metastasis and immunology, including cytokine-cytokine receptor interaction, cell adhesion molecules, antigen processing, and presentation, chemokine signaling pathway, natural killer cell mediated cytotoxicity, primary immunodeficiency and Th17 cell differentiation. The top 20 enriched pathways are presented in Fig 2F.

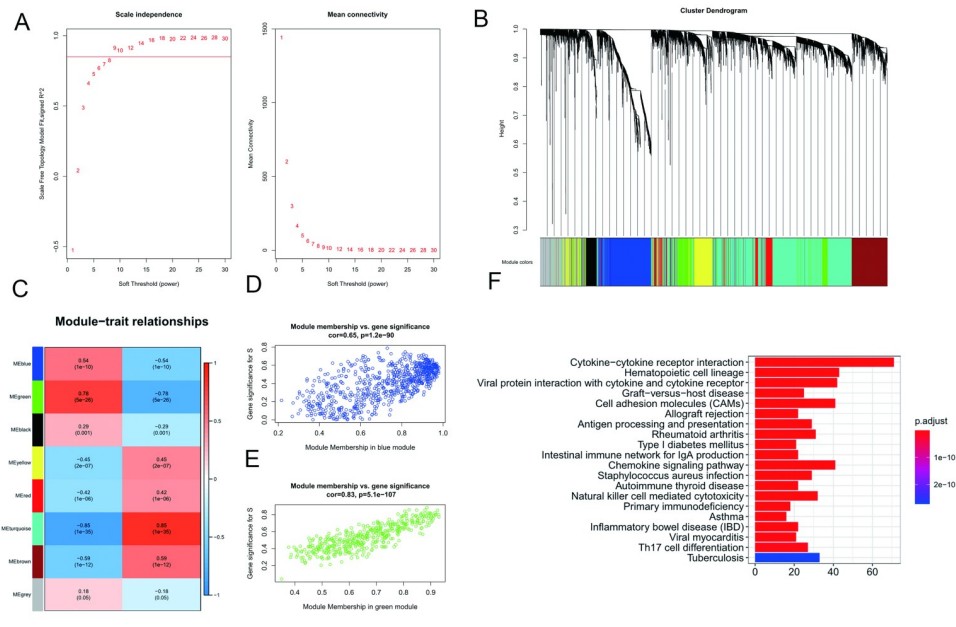

**Fig 2. Identification of hub seminoma-correlated module.** (A) Analysis of the scale-free fit index and mean connectivity for various soft-thresholding powers. (B) Cluster dendrogram of genes, with dissimilarity based on topological overlap. (C) Heatmap of the correlation between module eigengenes and clinical phenotypes. S, seminoma. NS, non-seminoma. Scatter plots of module eigengenes in the blue module (D) and green module (E). (F) Preliminary KEGG pathway enrichment analysis for genes in the blue module.

## Identification of DEGs in GSE8607

Under the cutoff criteria of Adjust P-value < 0.01 and |logFC| ≥2, 1297 DEGs were screened from the GSE8607 dataset. A heatmap and volcano map were plotted to show the DEGs (Fig 3A and 3B).

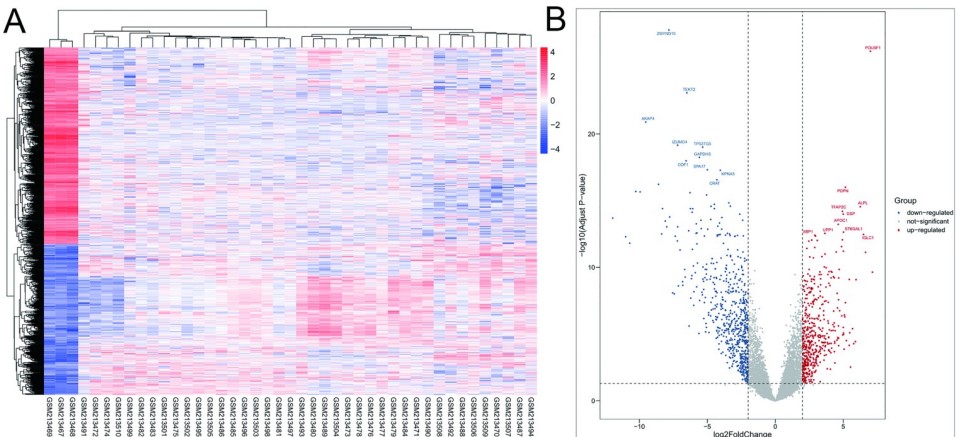

**Fig 3. Identification of DEGs in GSE8607.** (A) Heat map of the DEGs. (B) Volcano plot of the DEGs (cut-off criteria: adjust P-value < 0.01 and |logFC| ≥2).

## Identification of common genes, PPI network construction, and enrichment analysis

A total of 155 common genes were identified for further analysis, by application of the Venn-Diagram package in R (Fig 4A). Subsequently, an interaction network of 155 common genes was constructed and visualized using Cytoscape software. Then, the top 50 genes by highest BC were selected for enrichment analysis (Fig 4B). The results of biological process enrichment analysis revealed that the top 50 genes were mainly enriched in inflammation and immunity (Fig 4C). REACTOME Pathway enrichment analysis showed that the top 50 genes were mainly enriched in the immune system and signal transduction (Fig 4D).

## Investigation of the prognostic significance of hub genes

The top 10 genes by highest BC were identified in the PPI network and further analyses were performed on these candidate hub genes. According to Kaplan–Meier survival analyses, over-expression of *TYROBP* and *CD68* were significantly correlated with poor prognosis in seminoma (P < 0.05) (Fig 5A and 5B). The univariate Cox proportional hazards regression analyses showed that *CD68* and *ITGAM* were positively correlated with overall survival in seminoma (Table 1). The protein level of *ITGAM* was higher in seminoma tissues than in normal tissues (Fig 5C).

The expression levels of *TYROBP*, *CD68* and *ITGAM* in various tumors are presented in Fig 5D and 5E. We noticed that the expression values of these three genes were higher in TGCT compared to normal samples.

## Correlations of prognostic biomarkers with lymphocyte infiltration levels in seminoma

The enrichment analyses revealed that the top 50 genes were mainly enriched in immune-related pathways. Previous studies have demonstrated independent predictive roles for tumor-infiltrating lymphocyte grade in the survival of cancer and sentinel lymph node status [21]. Therefore, using the TIMER database, we further analyzed the correlations between the expression of the three candidate biomarkers and immune infiltrates in seminoma. *TYROBP*, *CD68*, and *ITGAM* expression had a significant positive correlation with infiltrating levels of B cells, CD4+ T cells, Macrophages, Neutrophils and Dendritic cells, as depicted in Fig 6.

Furthermore, the relationships between the expression of three prognostic biomarkers and immune marker genes for B cells, CD8+ T cells, neutrophils, macrophages, dendritic cells, NK cells, Th1 cells, Treg and monocytes, as reported in a previous study [20], were also explored in the TIMER database. The results demonstrated that most of the immune marker genes were significantly associated with *TYROBP*, *CD68* and *ITGAM* expression (Table 2).

These important findings further confirmed that the expression of the three prognostic biomarkers in seminoma was correlated with immune infiltration.

## Discussion

In recent decades, accumulated experiences and rapid development in surgeries, medicine, and radiology have provided an extremely high five-year survival rate for patients with TGCT, and this malignancy has become a kind of curable solid neoplasm [22]. However, for patients who fail first-line treatment or have solitary testicles, immunotherapy can be considered an alternative treatment strategy [5]. In such situations, it is necessary to identify novel biomarkers, which may be potential therapeutic targets and play critical roles in improving the prognosis of seminomas.

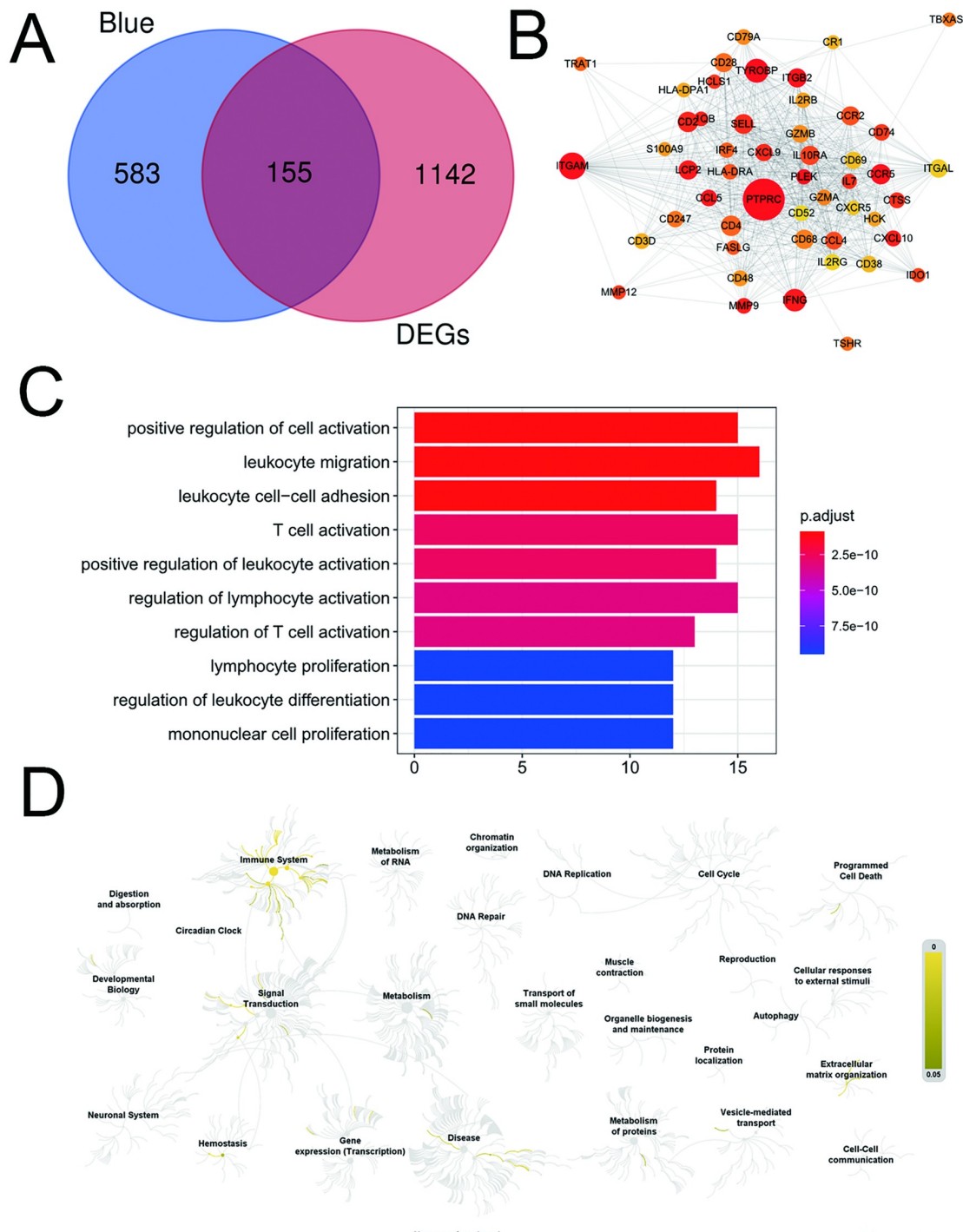

**Fig 4. Identification of hub biomarkers.** (A) Venn plot of common genes. (B) Top 50 genes by highest BC obtained from PPI network analysis. (C) Biological process enrichment analysis for the top 50 genes. (D) REACTOME Pathway enrichment analysis for the top 50 genes.

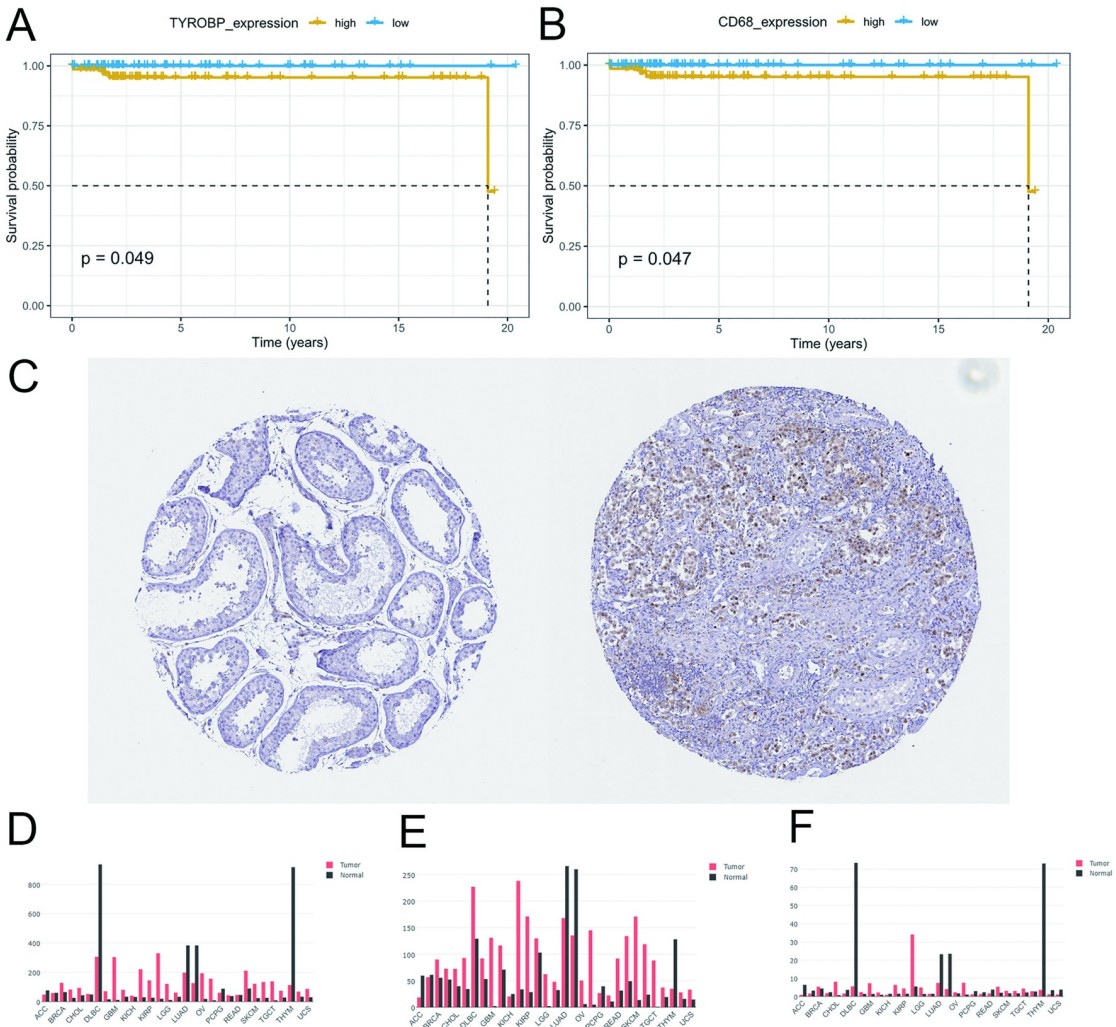

**Fig 5. Prognostic significance of hub biomarkers.** Kaplan–Meier survival analysis for *TYROBP* (A) and *CD68* (B). (C) Immunohistochemistry graph of *ITGAM* according to the Human Protein Atlas database (Reprinted from The Human Protein Atlas under a CC BY license, with permission from Inger Åhlén, original copyright August 28, 2020). Left: protein levels in normal tissues (staining: not detected, intensity: negative, quantity: none). Right: protein levels in seminoma tissues (staining: medium, intensity: moderate, quantity: > 75%). The expression level of *TYROBP* (D), *CD68* (E), and *ITGAM* (F) according to the GEPIA.

In the present study, the blue module with 738 genes was selected as the hub seminoma-correlated module. After intersection with 1297 DEGs in GSE8607, 155 common genes were identified for further analysis. Based on PPI network construction and analysis, the top 50 hub genes were screened, followed by functional enrichment analyses. The results revealed that the top 50 genes were mainly enriched in inflammation, immune response and signal transduction. Subsequently, the 10 genes with the highest BC were identified and evaluated by survival analyses and Cox hazards regression analyses, after which *TYROBP*, *CD68* and *ITGAM* were identified as prognostic biomarkers in seminoma. In addition, the correlation between these three biomarkers with immune infiltration implies an important role in tumor immunity in seminoma.

*TYROBP*, also known as *DAP12*, *KARAP* or *PLOSL*, encodes a transmembrane signaling polypeptide that contains an immunoreceptor tyrosine-based activation motif (*ITAM*) in its

**Table 1. The results of survival analyses and univariate cox analyses of the top 10 genes in the PPI network.**

| Gene symbol | Gene title | P value in Survival analysis | HR | P value in Univariate Cox analysis |
|---|---|---|---|---|
| CD68 | CD68 molecule | 0.047 | 3984.767 | 0.0467 |
| TYROBP | TYRO protein tyrosine kinase binding protein | 0.049 | 207.445 | 0.0858 |
| ITGAM | integrin subunit alpha M | 0.073 | 1494.224 | 0.0257 |
| PTPRC | protein tyrosine phosphatase, receptor type C | 0.079 | 169.022 | 0.1833 |
| IL10RA | interleukin 10 receptor subunit alpha | 0.082 | 3961.419 | 0.0840 |
| CCR5 | C-C motif chemokine receptor 5 (gene/pseudogene) | 0.090 | 603.833 | 0.1036 |
| SELL | selectin L | 0.195 | 17.488 | 0.2364 |
| CD4 | CD4 molecule | 0.195 | 897.657 | 0.0826 |
| IFNG | interferon gamma | 0.447 | 1.110 | 0.8818 |
| CD2 | CD2 molecule | 0.807 | 3.727 | 0.4816 |

HR, hazard ratio.

cytoplasmic domain. The encoded protein may associate with the killer cell immunoglobulin-like receptor (KIR) family of membrane glycoproteins and may act as an activating signal transduction element. That is to say that *TYROBP* plays critical roles in the immune system and signal transduction. Currently, its expression and clinical value have been studied in multiple cancers. Upregulated *TYROBP*, as previously reported by Stelios et al., was associated with advanced breast cancer grade and metastasis to the bone and liver [23]. Liu et al. conducted a bioinformatics analysis to identify biomarkers for liver cancer and found that *TYROBP* was the hub gene and may be a potential therapeutic target in liver cancer [24]. In addition, the results of a genome-wide cDNA microarray analysis showed that *TYROBP* was upregulated 5-fold or more in seminoma. The authors did not further explore the biological process for this gene in seminoma [25]. The overexpression of *TYROBP* in seminoma was also observed in our study and had predictive value for poor prognosis in patients with seminoma.

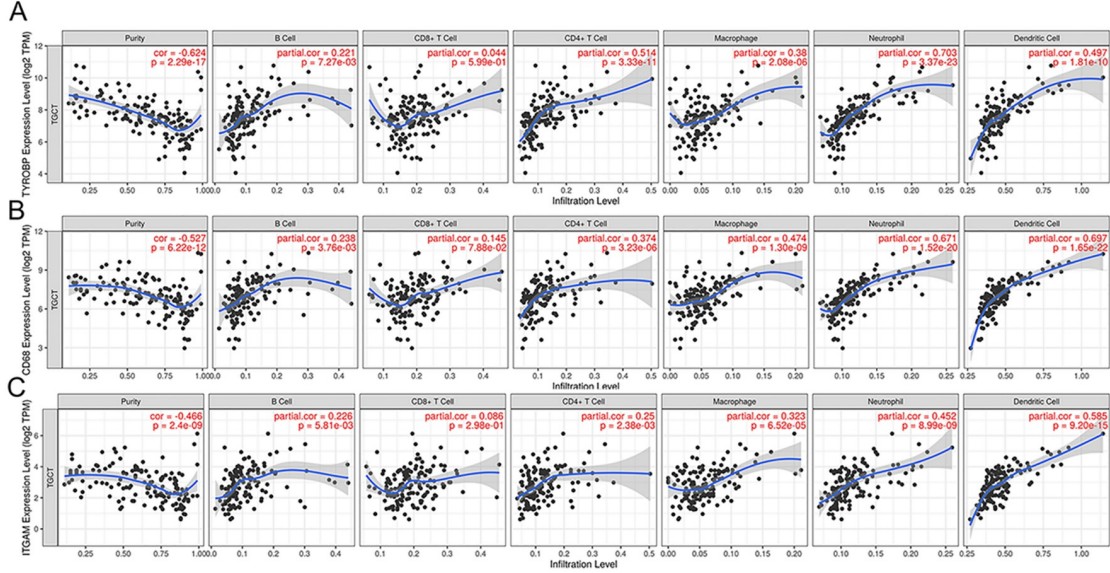

**Fig 6. Correlation between expressions of *TYROBP* (A), *CD68* (B) and *ITGAM* (C) and immune infiltration in seminoma according to the TIMER database.**

**Table 2. Correlation analysis between three prognostic biomarkers and immune cell type markers in the TIMER database.**

| Cell type | Gene markers | CD68 | | ITGAM | | TYROBP | |
|---|---|---|---|---|---|---|---|
| | | COR | P | COR | P | COR | P |
| B cells | FCRL2 | 0.536 | P < 0.01 | 0.406 | P < 0.01 | 0.606 | P < 0.01 |
| | CD19 | 0.393 | P < 0.01 | 0.262 | P < 0.01 | 0.467 | P < 0.01 |
| | MS4A1 | 0.473 | P < 0.01 | 0.358 | P < 0.01 | 0.544 | P < 0.01 |
| CD8+ T cells | CD8A | 0.639 | P < 0.01 | 0.470 | P < 0.01 | 0.671 | P < 0.01 |
| | CD8B | 0.621 | P < 0.01 | 0.386 | P < 0.01 | 0.623 | P < 0.01 |
| Neutrophils | FCGR3B | 0.212 | P < 0.01 | 0.096 | 0.243 | 0.162 | 0.048 |
| | CEACAM3 | 0.462 | P < 0.01 | 0.468 | P < 0.01 | 0.568 | P < 0.01 |
| | SIGLEC5 | 0.794 | P < 0.01 | 0.637 | P < 0.01 | 0.617 | P < 0.01 |
| | FPR1 | 0.680 | P < 0.01 | 0.559 | P < 0.01 | 0.564 | P < 0.01 |
| | CSF3R | 0.791 | P < 0.01 | 0.661 | P < 0.01 | 0.840 | P < 0.01 |
| | S100A12 | 0.182 | 0.026 | 0.136 | 0.098 | 0.212 | P < 0.01 |
| Macrophages | CD68 | 1.000 | P < 0.01 | 0.741 | P < 0.01 | 0.817 | P < 0.01 |
| | CD84 | 0.834 | P < 0.01 | 0.669 | P < 0.01 | 0.640 | P < 0.01 |
| | CD163 | 0.461 | P < 0.01 | 0.315 | P < 0.01 | 0.318 | P < 0.01 |
| | MS4A4A | 0.650 | P < 0.01 | 0.480 | P < 0.01 | 0.594 | P < 0.01 |
| Dendritic cells | CD209 | 0.490 | P < 0.01 | 0.399 | P < 0.01 | 0.412 | P < 0.01 |
| NK cells | KIR3DL3 | 0.383 | P < 0.01 | 0.249 | P < 0.01 | 0.367 | P < 0.01 |
| | NCR1 | 0.038 | 0.645 | 0.123 | 0.135 | -0.150 | 0.066 |
| Th1 cells | TBX21 | 0.681 | P < 0.01 | 0.522 | P < 0.01 | 0.738 | P < 0.01 |
| Treg | FOXP3 | 0.641 | P < 0.01 | 0.588 | P < 0.01 | 0.708 | P < 0.01 |
| | CCR8 | 0.296 | P < 0.01 | 0.404 | P < 0.01 | 0.088 | 0.283 |
| Monocyte | C3AR1 | 0.893 | P < 0.01 | 0.764 | P < 0.01 | 0.791 | P < 0.01 |
| | CD86 | 0.874 | P < 0.01 | 0.727 | P < 0.01 | 0.912 | P < 0.01 |
| | CSF1R | 0.865 | P < 0.01 | 0.689 | P < 0.01 | 0.784 | P < 0.01 |

NK cells, Natural killer cells; Th1 cells, type I helper T cells; Treg, regulatory T cells; COR, r value of Spearman's correlation.

Elena et al. reviewed relevant literature and concluded that *TYROBP* is a wiring component for NK cell anti-tumor function via its association with NKp44. In addition, *TYROBP* is associated with inflammation through its binding to specific receptors displayed by inflammatory cells such as monocytes/macrophages, neutrophils, and dendritic cells. Furthermore, the authors reported that *TYROBP* played essential roles in brain function and bone remodeling [26]. The roles of this gene seemed contradictory in the literature, due to its high expression in tumor samples and anti-tumor function through NK cell activation. We further searched the GEPIA to determine the expression of *TYROBP* in multiple cancers and normal samples. The results revealed that this gene was upregulated in most cancer samples, such as TGCT, breast cancer and cervical cancer, while in large B-cell lymphoma and thymoma, this gene was downregulated. Taken together with its important role in tumor immunity, this suggests the gene may be cancer-specific and its aberrant expression is correlated with tumorigenesis.

*CD68*, as reported in a previous study, provided a good predictive value as a prognostic marker for survival in cancer patients. The authors described that low expression of this gene was found in tumor cells [27]. After consulting the GEPIA, we noticed that the gene was upregulated in most cancers including TGCT and downregulated in lung adenocarcinoma and thymoma.

The expression and effects of *CD68* have mostly investigated in immunohistochemical studies of various tumors. For example, positive immunophenotypical features of *CD68* have

been observed in bellini carcinoma, a rare type of renal malignancy [28], and testicular myeloid sarcoma [29]. Regarding the features of *CD68* in seminoma, Tine et al. studied the phenotypic characterization of immune cell infiltrates in 41 TGCTs and found that a high proportion of them were identified as *CD68*+ macrophages [30]. Moreover, the authors reported the absence of active immune surveillance in TGCT, suggesting a potential role for *CD68* in tumor immunity. Sam et al. performed immunohistochemistry in 51 seminomas and 26 non-seminomatous germ cell tumors, and found that germ cell tumors primarily expressed PD-L1 (a known checkpoint in tumor immunity) on tumor-associated *CD68*+ macrophages [31]. Furthermore, the expression features of these macrophages were more significant in seminomas than in non-seminomatous germ cell tumors. These results provide robust evidence that *CD68* is a key molecule in the pathological process of seminoma. Additionally, we speculate that the gene may have a potential association with immune checkpoint pathways according to the findings provided in the published literature.

*ITGAM*, also known as macrophage-1 antigen (*Mac-1*) or complement receptor 3 (*CR3*), has been explored in multiple types of diseases. Numerous previous studies have reported a biological function for *ITGAM* in the development of systemic lupus erythematosus [32, 33]. Agarwal et al. conducted proteomic analysis to identify core sperm proteins in patients with seminoma via cryopreserved semen samples [34]. The results revealed that *ITGAM* protein was downregulated in seminoma, and may be involved in spermatogenesis, motility function, and infertility. The potential mechanism for *ITGAM*-relevant asthenozoospermia in patients with testicular cancer was also studied by Selvam et al. [35]. However, direct evidence on *ITGAM* and its molecular mechanism in seminoma are limited in the present literature.

Moreover, potential roles for *ITGAM* in various malignant tumors have also been reported. One study by Joanna et al. explored *ITGAM* in the progression and prognosis of renal cancer and found that aberrant expression of *ITGAM* was significantly correlated with renal cell carcinoma as compared with controls. Moreover, the expression signature of this gene was strongly associated with poor survival [36]. *ITGAM* and *ITGB6* have been confirmed to play critical roles in ovarian cancer invasion and implant metastasis [37]. One study investigating biomarkers in breast cancer brain metastasis via integrated genomic and epigenomic analysis showed that hypermethylation and downregulation of *ITGAM* were associated with defects in cell migration and adhesion [35]. One meeting report in 2016 described that *ITGAM* protein positive tumor associated macrophages were associated with tumor angiogenesis promotion and immunosuppression [38].

Nevertheless, the limitations of this study must be clearly pointed out. First, future studies in vivo or in vitro are needed to elucidate the detailed molecular mechanisms for these hub genes in seminoma. Second, a larger number of samples are required to make our findings more convincing. Third, in our study, three prognostic biomarkers were identified and analyzed. Zaman and colleagues also performed an integrated network analysis by integrating genomic alteration information and functional genetic data. They found that the networks could effectively predict subtype-specific drug targets which have been experimentally validated. By taking advantage of this integrated network analysis, more immunotherapy targets for seminoma could be identified and clinically applied [39]. Emerging evidence has shown that non-coding RNA biomarkers play important roles in various human diseases including seminoma [40]. With the rapid development of computational prediction models, Chen et al. proposed several innovative prediction models to identify non-coding RNA biomarkers correlated with human diseases [41–44]. Future studies will attempt to find significant non-coding RNA biomarkers of seminoma and may take advantage of these state-of-the-art computational models.

## Conclusion

Three novel biomarkers, *TYROBP*, *CD68* and *ITGAM*, were identified from databases and correlated with poor prognosis in patients with seminoma. Furthermore, all of them were significantly positively correlated with immune infiltration, indicating that they may be potential targets for immunotherapy. Future experimental studies are needed to validate our findings and explore the molecular mechanisms of the three genes in the context of seminoma.

## Supporting information

**S1 Fig. Flowchart of the construction of weighted correlation network.**
(TIF)

## Author Contributions

**Conceptualization:** Hualin Chen, Gang Chen.

**Data curation:** Hualin Chen.

**Formal analysis:** Hualin Chen.

**Funding acquisition:** Gang Chen.

**Investigation:** Hualin Chen, Xiaoxiang Jin.

**Methodology:** Hualin Chen, Xiaoxiang Jin.

**Project administration:** Hualin Chen, Xiaoxiang Jin.

**Resources:** Hualin Chen, Xiaoxiang Jin.

**Software:** Hualin Chen, Yang Pan, Xiaoxiang Jin.

**Supervision:** Hualin Chen, Yang Pan.

**Validation:** Hualin Chen, Yang Pan.

**Visualization:** Hualin Chen, Yang Pan.

**Writing – original draft:** Hualin Chen.

**Writing – review & editing:** Hualin Chen, Gang Chen.

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
