## [Decision Letter · Decision Letter 0]

13 Aug 2020

PONE-D-20-17839

Three novel prognostic biomarkers for seminoma identified by weighted gene coexpression network analysis

PLOS ONE

Dear Dr. Chen,

Thank you for submitting your manuscript to PLOS ONE. After careful consideration, we feel that it has merit but does not fully meet PLOS ONE’s publication criteria as it currently stands. Therefore, we invite you to submit a revised version of the manuscript that addresses the points raised during the review process.

We look forward to receiving your revised manuscript.

Kind regards,

Edwin Wang

Academic Editor

PLOS ONE

Journal Requirements:

2.  We note that Figure 5C in your submission contain copyrighted images. All PLOS content is published under the Creative Commons Attribution License (CC BY 4.0), which means that the manuscript, images, and Supporting Information files will be freely available online, and any third party is permitted to access, download, copy, distribute, and use these materials in any way, even commercially, with proper attribution. For more information, see our copyright guidelines: http://journals.plos.org/plosone/s/licenses-and-copyright.

a)     You may seek permission from the original copyright holder of Figure(s) [#] to publish the content specifically under the CC BY 4.0 license.

Additional Editor Comments (if provided):

A related work using network modules (PMID: 24075989) should be discussed

Reviewers' comments:

Reviewer's Responses to Questions

**Comments to the Author**

1. Is the manuscript technically sound, and do the data support the conclusions?

Reviewer #1: No

Reviewer #2: Yes

2. Has the statistical analysis been performed appropriately and rigorously? 

Reviewer #1: No

Reviewer #2: Yes

3. Have the authors made all data underlying the findings in their manuscript fully available?

Reviewer #1: Yes

Reviewer #2: Yes

4. Is the manuscript presented in an intelligible fashion and written in standard English?

Reviewer #1: No

Reviewer #2: No

5. Review Comments to the Author

Reviewer #1: The present manuscript titled as “Three novel prognostic biomarkers for seminoma identified by weighted gene coexpression network analysis” used a WGCNA method to analyze seminoma-correlated modules and core genes for identifying prognostic biomarkers for seminoma. Overall, the approach employed in this study is to straightforward and the results are not solid. Below are some comments:

1. “Three novel prognostic biomarkers” is not clear. I suggest to revise it as “Three-gene prognostic biomarkers”.

2. Abstract is not concise enough.

3. Many grammatic errors. English should be improved.

4. Table 1: Why “P value in Survival analysis” and “P value in Univariate Cox analysis” were different? The “P value in Univariate Cox analysis” for TYROBP is not less than 0.05, and the “P value in Survival analysis” for ITGAM is also not less than 0.05. So how to interpret these genes were significantly corelated with survival?

5. The authors claimed the three genes can be used as prognostic biomarker, but no intendent dataset was used for validation. They just analyzed the association of these genes to survival in the TCGA dataset.

Reviewer #2: The author used weighted gene coexpression network analysis and identified three novel prognostic biomarkers for seminoma. I hope the manuscript could be further strengthened by the following comments.

1. Please clearly state the major innovation of this work.

2. More details about the construction of weighted correlation network should be given. How was the correlation matrix constructed and transformed into weighted adjacency matrix? What is the blue and green modules?

3. Please add a flowchart of the construction of weighted correlation network.

4. Please give the reason for parameter selection (e.g., the soft threshold).

5. I want to know whether this network analysis can also be used for the analysis of the other cancers?

6. You should revise your English writing carefully and eliminate small errors in the paper to make the paper easier to understand.

7. Could you discuss the recent trend of developing computational model for identification of the non-coding RNA biomarker of human complex diseases as the future direction of your current research about biomarker identification of seminoma? Some important studies should be cited and discussed (PMIDs: 31927572, 29939227, 29045685, and 24002109).

6. PLOS authors have the option to publish the peer review history of their article (what does this mean?). If published, this will include your full peer review and any attached files.

Reviewer #1: No

Reviewer #2: No

---

## [Author Response · Author response to Decision Letter 0]

15 Sep 2020

Dear editors,

Re: Manuscript reference PONE-D-20-17839

 Please find attached a revised version of our manuscript “Three-gene prognostic biomarkers for seminoma identified by weighted gene coexpression network analysis”, which we would like to resubmit for publication in PLOS ONE. 

 The comments of the academic editor and reviewers were highly insightful and enabled us to greatly improve the quality of our manuscript. In the following pages are our point-by-point responses to each of the comments.

 Revisions in the text are shown using the red highlight for additions, and strikethrough font [example: Three novel] for deletions. The revisions have been organized according to the journal requirements. Written permission from the copyright holder to publish Figure 5C under the CC BY 4.0 license is obtained and uploaded simultaneously. The specific text was added to the copyrighted figure caption in the revision. We have sought for English-language editing service for help, considering many grammatical errors in the original manuscript. And the service proof is uploaded with the revisions. 

We look forward to hearing from you regarding our submission. We would be glad to respond to any further questions and comments that you may have.

 We shall look forward to hearing from you at your earliest convenience.

Thank you and best regards.

Yours sincerely,

Hualin Chen

Corresponding author:

Name: Gang Chen

E-mail: chengang2308@163.com

Response to the comment of Academic Editor

1. A related work using network modules (PMID: 24075989) should be discussed

Response: Thank you for your comment and this nice work has been discussed in the revision. Zaman et al performed wonderful integrated network analysis and provided a unique insight into the underlying mechanisms of cancer cell survival and proliferation driven by genomic alterations. More importantly, drug targets could be identified through this network analysis, indicating the potential significant clinical value of this work. 

Responses to the comments of Reviewer #1

1. “Three novel prognostic biomarkers” is not clear. I suggest to revise it as “Three-gene prognostic biomarkers”. 

Response: Thanks for your advice and the title is revised. 

2. Abstract is not concise enough.

Response: Thank you for your comment and the abstract is revised.

3. Many grammatic errors. English should be improved.

Response: I feel ashamed of my poor English and I am sorry for your unpleasant reading experience. I have asked the English language-editing service for help to revise the paper. 

4. Table 1: Why “P value in Survival analysis” and “P value in Univariate Cox analysis” were different? The “P value in Univariate Cox analysis” for TYROBP is not less than 0.05, and the “P value in Survival analysis” for ITGAM is also not less than 0.05. So how to interpret these genes were significantly corelated with survival?

Response: Thanks for your comments. (1) “P-value in Survival analysis” indicated the P value calculated by log-rank test (Kaplan–Meier survival analysis), while “P-value in Univariate Cox analysis” indicated the P-value calculated by Cox proportional hazards regression model. Differences in statistical principles may cause different results. However, both methods were widely used in prognostic biomarkers identification. (2) The overall survival rate of seminoma patients was relatively high and fewer death events were observed in the TCGA datasets. If only one method was employed, some clinical valuable biomarkers may be ignored. For example, according to the KM survival analysis (log-rank test), gene ITGAM was not significantly correlated with prognosis. While univariate cox analysis and immunohistochemistry suggested the potential prognostic value of this biomarker. With the attempt to identify more valuable biomarkers of seminoma, we think that it may be a proper manner to integrate the results of both methods, instead of the intersection. (3) As we addressed in our paper, the lack of further experimental validation was the main drawback of our work. Therefore, further researches are urgently needed to validate our findings and explore the molecular mechanism of these biomarkers with tumor immunity in-depth. 

5. The authors claimed the three genes can be used as prognostic biomarker, but no intendent dataset was used for validation. They just analyzed the association of these genes to survival in the TCGA dataset.

Response: I am sorry for the limitation of our work. Seminoma is rare in the general population and the health testis is much hard to obtain. Lack of tumor samples prevents us from further validation. Unfortunately, other common electronic tumor databases like the International Cancer Genome Consortium (ICGC) do not provide seminoma cases for analysis. However, the three biomarkers were also DEGs in GSE8607. 

I appreciate your great efforts in our work and your precious comments help us improve the quality of our manuscript. 

Responses to the comments of Reviewer #2

1. Please clearly state the major innovation of this work.

Response: (1) This is the first study using WGCNA to identify prognostic biomarkers of seminoma and three prognostic biomarkers, which may be potential immunotherapy targets, are obtained; (2) Tumor immunity is a hot topic currently and previous studies have mentioned the critical role of the immune response in seminoma patients. Our findings further prove the roles of tumor immune in seminoma, which may provide evidence for future immunotherapy.

2. More details about the construction of weighted correlation network should be given. How was the correlation matrix constructed and transformed into weighted adjacency matrix? What is the blue and green modules?

Response: Thanks for your suggestions. The official tutorials (available at https://horvath.genetics.ucla.edu/html/CoexpressionNetwork/Rpackages/WGCNA/) provide the codes for analysis in detail. What we have to do is to adjust some thresholds according to our samples. (1) More details about the construction of the weighted correlation network are given in the revision. (2) The correlation matrix is constructed based on Pearson correlation analysis (more details are given in the revision). Then, a soft‑thresholding function is used to transform the correlation matrix into a weighted and scale-free adjacency matrix. In this part, identification of a soft-thresholding β (a soft-thresholding parameter enhancing strong correlations between genes and penalizing weak correlations) is of importance. (3) The blue and green modules represent different modules, and the colors of the modules are just used to mark and distinguish these different modules. The colors have no relation to the genes. Genes in the same module have strong correlation with each.

3. Please add a flowchart of the construction of weighted correlation network.

Response: Thank you for your advice. Actually, Figure 2 A-E represents the main workflow of the whole WGCNA. However, we generate a flowchart of the construction of a weighted correlation network according to your suggestion and it is uploaded as a supplementary figure. 

4. Please give the reason for parameter selection (e.g., the soft threshold).

Response: Thanks for your comment. (1) Soft threshold: a suitable soft threshold is determined by two factors: scale-independence and connectivity. In our study, scale-independence >0.85 with maximum average connectivity is considered as the criteria for soft threshold selection. And as Figure 2A demonstrated, only power 9 meets the criteria and is selected for further analysis. As reported in the official guidelines, scale-independence >0.80 is acceptable in soft threshold selection. Under certain special conditions, it is less likely to obtain proper power. And power 6 is often empirically used. (2) Adjust P-value < 0.01 and |logFC| ≥2: Actually, both 0.01 and 0.05 can be used as the thresholds of Adjust P-value. As for |logFC|, 1, 2 or even 0.5 is acceptable. There is no strict restriction in the thresholds of Adjust P-value and |logFC|. In our study, the reason we choose a stricter threshold is that a large number of DEGs in GSE8607 are identified through initial analysis and we only want to find out more significant genes with lower Adjust P-value and higher |logFC|. 

5. I want to know whether this network analysis can also be used for the analysis of the other cancers?

Response: Thanks for your question and the answer is yes.

6. You should revise your English writing carefully and eliminate small errors in the paper to make the paper easier to understand.

Response: I am extremely sorry for the grammar errors in our original paper. I have sought for language-editing service for help. 

7. Could you discuss the recent trend of developing computational model for identification of the non-coding RNA biomarker of human complex diseases as the future direction of your current research about biomarker identification of seminoma? Some important studies should be cited and discussed (PMIDs: 31927572, 29939227, 29045685, and 24002109).

Response: Your comments and advice are filled with wisdom. These researches contribute a lot to the development of computational biology and these computational models provide new methods to find novel and more significant disease-related non-coding RNA biomarkers. More excellent works, I believe, could be carried out soon based on these models. These works have been cited and discussed in the revision.

---

## [Decision Letter · Decision Letter 1]

6 Oct 2020

Three-gene prognostic biomarkers for seminoma identified by weighted gene co-expression network analysis

PONE-D-20-17839R1

Dear Dr. Chen,

We’re pleased to inform you that your manuscript has been judged scientifically suitable for publication and will be formally accepted for publication once it meets all outstanding technical requirements.

Kind regards,

Edwin Wang

Academic Editor

PLOS ONE

Additional Editor Comments (optional):

Reviewers' comments:

Reviewer's Responses to Questions

**Comments to the Author**

1. If the authors have adequately addressed your comments raised in a previous round of review and you feel that this manuscript is now acceptable for publication, you may indicate that here to bypass the “Comments to the Author” section, enter your conflict of interest statement in the “Confidential to Editor” section, and submit your "Accept" recommendation.

Reviewer #1: All comments have been addressed

Reviewer #2: All comments have been addressed

2. Is the manuscript technically sound, and do the data support the conclusions?

Reviewer #1: Yes

Reviewer #2: Yes

3. Has the statistical analysis been performed appropriately and rigorously? 

Reviewer #1: Yes

Reviewer #2: Yes

4. Have the authors made all data underlying the findings in their manuscript fully available?

Reviewer #1: Yes

Reviewer #2: Yes

5. Is the manuscript presented in an intelligible fashion and written in standard English?

Reviewer #1: No

Reviewer #2: Yes

6. Review Comments to the Author

Reviewer #1: Please provide the source code of the analysis in this study publically available, which is necessary for reproducing the results.

Reviewer #2: It could be accepted now

7. PLOS authors have the option to publish the peer review history of their article (what does this mean?). If published, this will include your full peer review and any attached files.

Reviewer #1: No

Reviewer #2: No

---

## [Editor Report · Acceptance letter]

16 Oct 2020

PONE-D-20-17839R1 

Three-gene prognostic biomarkers for seminoma identified by weighted gene co-expression network analysis 

Dear Dr. Chen:

I'm pleased to inform you that your manuscript has been deemed suitable for publication in PLOS ONE. Congratulations! Your manuscript is now with our production department. 

Kind regards, 

on behalf of

Dr. Edwin Wang 

Academic Editor

PLOS ONE